# Effect of Mechanical Activation on the Leaching Process of Rare Earth Metal Yttrium in Deep Eutectic Solvents

**Xiaofen Li** [1,2], **Wei Li** [2], **Yuntao Gao** [2] and **Guocai Tian** [1,*]

1   State Key Laboratory of Complex Non-Ferrous Metal Resource Clean Utilization, Faculty of Metallurgical and Energy Engineering, Kunming University of Science and Technology, Kunming 650093, China
2   School of Chemistry and Environment, Yunnan Minzu University, Kunming 650500, China
*   Correspondence: tiangc01@gmail.com; Tel.: +86-137-5955-8375

**Abstract:** Deep eutectic solvents (DESs) have the potential to be a lixiviant for the selective processing of metal because of their versatile complexation properties. In this study, the leaching behavior of rare-earth carbonate before and after mechanical activation in chloride–urea–malonic acid, the deep eutectic solvents (ChCl-urea-MA DESs) were investigated. Leaching experiments were employed to investigate the effects of reaction temperature and activation time on the leaching efficiency of the metal, yttrium, in DESs. The leaching efficiency of yttrium was determined to be 85.2% when the activation time was 60 min, the leaching temperature was 80 °C and the leaching time was 12 h. The findings showed that mechanical activation increased the yttrium leaching efficiency from 48.61 to 88.37% by lowering the particle size and increasing the contact area of the reaction. The investigation of the yttrium leaching kinetics revealed that, after mechanical grinding, the apparent activation energy of rare-earth carbonate decreased from 83.88 kJ·mol$^{-1}$ to 37.4 kJ·mol$^{-1}$, and the leaching process of the sample changed from controlled by chemical reaction to controlled by diffusion in the solid product layer. Mechanical activation combined with DESs was proposed as an eco-friendly, sustainable, and effective alternative to conventional mineral acid leaching and solvent, with advantages such as moderate conditions, reusability of the DESs, and no additional wastewater produced. The findings of the study show this method is a good way to recycle rare-earth metals.

**Keywords:** rare-earth carbonate; rare-earth metal; mechanical activation; deep eutectic solvents

## 1. Introduction

Rare earth metals play a vital role in numerous fields of high-tech industry, including the magnetic, electrical, and catalytic industries, due to their remarkable chemical and physical properties [1]. Carbonates and phosphates are the predominant minerals that are commercially processed for rare-earth metal extraction [2]. In general, it is difficult to dissolve metal oxides in molecular solvents; hence, strong mineral acid solutions, such as nitric or sulfuric acid, are typically used for this purpose [3]. However, the use of common organic solvents remains an issue of economic and environmental concern [4]. Consequently, the generation of diverse forms of waste material during metal separation/extraction with variable degrees of negative environmental impacts compels the mineral industry to develop new environmentally viable approaches, without compromising on cost competitiveness and cost effectiveness.

Because of their long life span, increased luminous efficiency, and lower energy consumption, fluorescent lamps have increasingly replaced incandescent lamps and are extensively used in lighting systems around the world, owing to the growing emphasis on green and low-carbon economies [5]. According to the statistics, a considerable number of waste rare-earth fluorescent lamps are processed as solid waste each year, which not only results in the waste of rare-earth resources but also poses a serious threat to the environment [6]. Recycling fluorescent lamps for the rare-earth elements has significant

practical implications. The phosphors from waste lamps can be recycled by several different processes, such as acid leaching [7], solvent extraction separation [8], extraction of supercritical fluid [9], alkaline fusion [10], and mechanical activation [11]. However, these recovery technologies have several drawbacks, including low leaching efficiency, high energy consumption, and harsh environmental conditions.

Deep eutectic solvents (DESs) were defined as an extended class of ILs which are known to be environmentally friendly compounds [12]. DESs are systems formed from the eutectic mixtures of Lewis or Brønsted acids and bases, and the mixture of choline chloride with urea as the hydrogen-bond donor (HBD) is a representative example [13]. Even under very moderate conditions, DESs have a significant solvation potential for the hardly soluble metal oxides, indicating their usefulness for treating metal oxides. Researchers are increasingly interested in DES applications in metallurgy, since their production costs are comparable to traditional solvents while exhibiting great chemical stability, nontoxicity, and biodegradability [14,15].

Mechanical activation is one of the most essential strategies for improving hydrometallurgical processes. It has been effectively implemented in mineral processing, materials engineering, and chemical engineering [16,17], and demonstrates excellent application potential for extracting metals from a variety of waste types. Moreover, it has been found that it considerably increases the solubility rate of minerals [18,19]. It leads to physical disintegration and the development of active surfaces, as well as alterations in the physical and chemical characteristics of the particles, including a reduction in the particle size, an increase in the specific surface area, a crystalline structural deformation, and the formation of new phases that are more susceptible to leaching [20–22].

In the present investigation, rare-earth metal leaching of pure yttrium carbonate salts was investigated, utilizing ChCl-based DESs in order to reveal the behavior of rare-earth metal leaching. Moreover, the influence of the mechanical activation on the leaching process of the rare-earth metal, yttrium, in DESs was examined. This study's findings not only show a simple way for extracting rare-earth metals but also provide a theoretical background and substantial direction for the leaching of rare-earth metals from waste fluorescent lamps. In this investigation, an environmentally friendly mechanical activation based on DESs was established as a pretreatment process technique that can effectively disrupt the structure of the rare-earth carbonate and enhance the recovery efficiency of yttrium. The influences of activation on the rare-earth metal leaching efficiency were explored, and optimal conditions were discovered.

## 2. Experimental

### 2.1. Materials and Methods

The choline chloride (ChCl) was purchased from Shanghai Aladdin Bio-Chem Technology Co., LTD (Shanghai, China) (>98%) with 99% purity. Prior to usage, it was recrystallized from 100% ethanol and dried overnight (90 °C, −0.08 Mpa) under a vacuum. The malonic acid (MA) and urea (Ur), were acquired from Sigma-Aldrich (>99%) and were allowed to dry overnight before use (>99%). The rare-earth yttrium carbonate was purchased from Shenzhen Zhanzhanlong Technology Co., Ltd. (Shenzhen, China) (>94%). The arsenazo-III, anhydrous sodium acetate, anhydrous ethanol, and glacial acetic acid, were purchased from Shenzhen Zhanzhanlong Technology Co. Ltd. All of the reagents were of analytical grade.

The DESs were obtained by mixing ChCl in a 1:1:0.5 molar ratio with HBDs (UR and MA) and stirring the mixture at 90 °C until a homogenous liquid was achieved. The resultant DESs were packed and stored at 80 °C before being utilized immediately without any further purification.

### 2.2. Equipment and Characterization

Scanning electron microscopy (SEM) was employed to assess the morphology as well as the size distribution. The sample was ground, using a SCINTZ-48 high-throughput tissue grinder (Ningbo Xinzhi Biotechnology Co., Ltd., Ningbo, China). A TU-1950 double-beam

UV-Vis spectrophotometer (Beijing Puxi General Instrument Co., Ltd., Beijing, China) was used to determine the metal content dissolved in the DESs. A TDL-60B low-speed desktop centrifuge (Shanghai Anting Scientific Co. Ltd., Shanghai, China) was used to separate the rare-earth yttrium carbonate from the DESs, following the leaching experiments.

### 2.3. Mechanical Activation Operation

The mechanical activation of the rare earth yttrium carbonate was completed, using a SCINTZ-48 high-throughput tissue grinder. In each activation batch, the yttrium carbonate powder (1.0 g) was weighed and then mixed with a steel grinding ball (3 mm in diameter) in a ball-milling jar with an inner diameter of 10 mm and an inner volume of 15 mL, then it was mechanically activated at 1800 r/min for a different time. During the activation operation, the mill was set to run and pause alternatively at a 15 min interval to avoid the accumulation of generated heat. All of the activated samples were subjected to DESS leaching within 12 h after mechanical activation.

### 2.4. Leaching Experiments

Small centrifuge tubes (5.0 mL) were filled with a constant amount of DESs (4.0 g), 0.5 mg sample, and a small amount of water (7.5 wt.%) during the various experiments. All of the amounts were presented per gram of DESs (mg·g$^{-1}$). The centrifuge tubes were placed in a constant temperature mixer, and the leaching experiments were carried out at a set temperature while shaken at 800 r/min over a specified time. The undissolved material was then collected, and the clear DESs were obtained by centrifuging the tubes (5500 rpm, 30 min). UV-Vis spectrophotometry was employed to assess the metal concentration dissolved in the DESs, and the leaching efficiency was calculated. The UV-visible absorption spectra are shown in Figure 1.

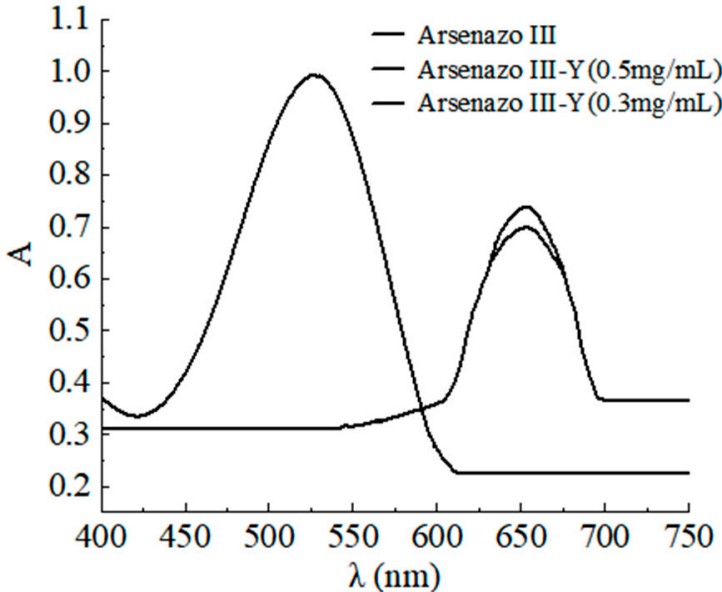

**Figure 1.** UV-visible absorption spectra.

It can be seen that the complex formed by arsenazo-III and the rare-earth metal has a characteristic absorption peak at 653 nm. There is an absorption peak at 520 nm for the arsenazo-III without the rare-earth metal, while the absorption peak of arsenazo-III shifted from 525 nm to 653 nm after adding the rare-earth elements. Then, the absorption peaks at 653 nm increased with the increase in the rare-earth content. Therefore, the absorbance was measured at the wavelength of 653 nm.

## 3. Results and Discussion

### 3.1. The Solubility of Yttrium Carbonate in DES$_S$ before and after Mechanical Activation

The yttrium carbonate sample before (a) and after (b) being mechanically activated was leached in DESs for 12 h and then centrifuged at 1800 r/min for 10 min. As shown in Figure 2a, a large amount of the sample was deposited at the bottom of the centrifuge tube, which was barely dissolved. Meanwhile, there was almost no sample at the bottom, which had almost completely dissolved, as illustrated in Figure 2b. The findings demonstrated that the activated yttrium carbonate is more soluble in DESs.

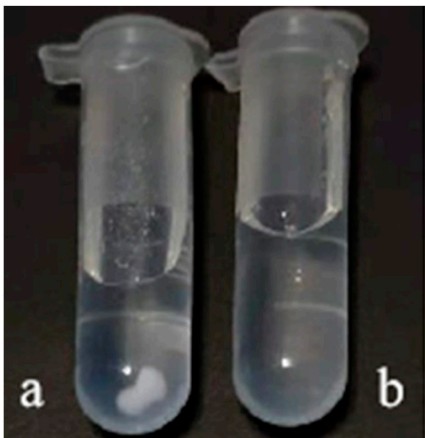

**Figure 2.** The solubility of yttrium carbonate in DES$_S$ and before (**a**) and after (**b**) mechanical activation.

### 3.2. SEM Analysis of Yttrium Carbonate before and after Mechanical Activation

The micromorphology analysis of yttrium carbonate before and after being mechanically activated was characterized by SEM micrographs of milled and un-milled samples and is presented in Figure 3. As depicted in Figure 3a, the un-milled samples were comprised of irregular particles of varying sizes with compact surfaces, and the inactivated samples had higher particle diameters, as well as compact and massive agglomeration. However, the particle diameter decreased obviously and dispersed homogeneously after mechanical activation, as illustrated in Figure 3b. The results demonstrated that mechanical activation can promote particle disintegration and the development of new surfaces.

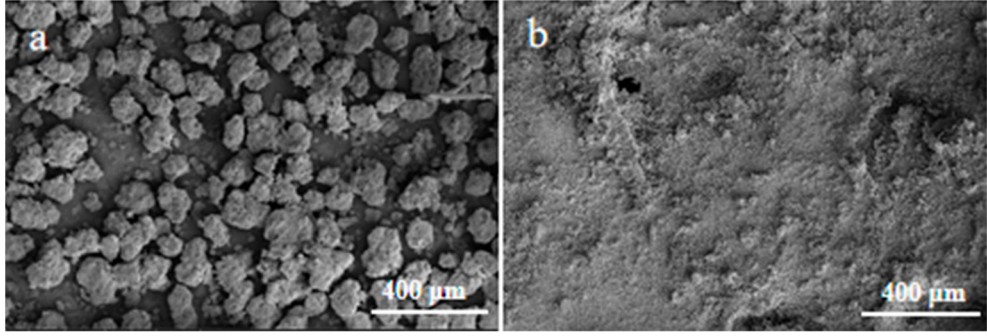

**Figure 3.** SEM images of the inactivated sample (**a**) and mechanically activated sample (**b**).

### 3.3. Effect of Mechanical Activation Time on the Leaching Efficiency of Metallic Yttrium

The effect of the different mechanical activating times on the leaching efficiency of yttrium metal in yttrium carbonate was investigated under the reaction temperature of 80 °C. As shown in Figure 4, the leaching efficiency of the metallic yttrium in DESS increased as the mechanical activating time increased from 20 to 90 min. When the mechanical activating time reached 60 min, the leaching efficiency of the metallic yttrium tended to be flat, and was independent of mechanical activating time beyond 60 min.

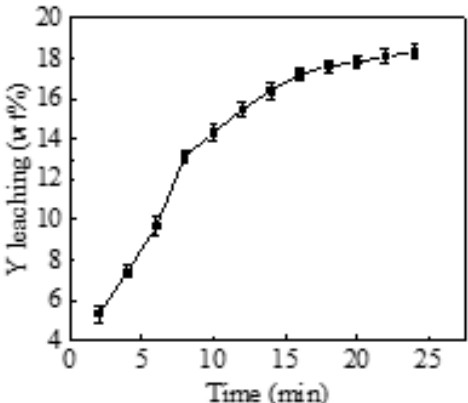

**Figure 4.** The effects of mechanical activating time on leaching efficiency (T = 80 °C).

The reason for this is that mechanical activation breaks down the agglomerates, causing the particle size to decrease, and increasing the corresponding specific surface area. However, when the particles reached a certain degree of refinement, the local plastic deformation and mutual penetration of particles in the contact area of adjacent particles were induced due to the significant increase in Van der Waals force. Then the particles began to adhere and aggregate, which resulted in an increase in the particle size and a gradual decrease in the specific surface area, and eventually no longer changed with mechanical activating time. Therefore, 60 min was chosen as the optimal activation time.

### 3.4. Effect of Mechanical Activation on the Leaching Process

Under the condition of the reaction temperature of 80 °C, and the mechanical activation time ($t_{MA}$) of 60 min, the effect of mechanical activation on the leaching efficiency of yttrium was investigated, and, as depicted in Figure 5, the sample's mechanical activation treatment can significantly improve the leaching efficiency of Ye. The leaching efficiency of the inactivated sample in DESs increased gradually with time; when the leaching period reached 8 h, the leaching efficiency tended to stabilize, and the maximum leaching efficiency was 46.5%. The leaching efficiency of the activated sample, on the other hand, increased rapidly with the increasing reaction time, and then tended to be moderate when the leaching period reached 12 h, with a maximum leaching efficiency of 88.37%. The maximum leaching efficiency of the samples before and after mechanical activation substantially doubled, from 48.61 to 88.37% in the same time. This is attributed to breaking down the agglomerates and increases in the surface area caused by mechanical activation. Therefore, mechanical activation can significantly increase the yttrium leaching efficiency in the sample.

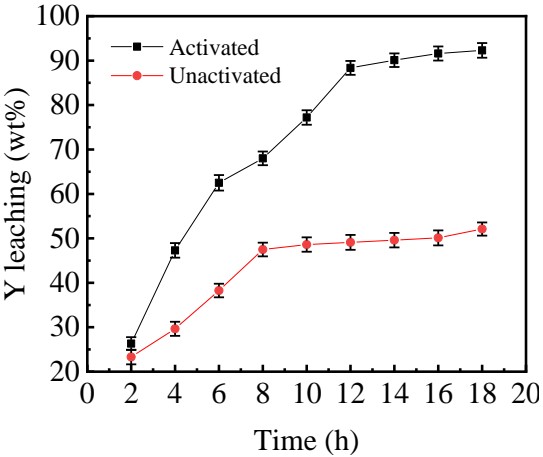

**Figure 5.** The effect of mechanical activation and leaching time on leaching efficiency (T = 80 °C, $t_{MA}$ = 60 min).

### 3.5. Effect of Reaction Temperature on the Leaching Process

To explore the influence of temperature, rare-earth carbonate was leached at various temperatures, ranging from 50 to 80 °C under the condition of the mechanical activation time of 60 min, and the leaching time ($t_L$) of 12 h; the findings are displayed in Figure 6. Figure 6 shows that the leaching efficiency of yttrium increases from 20.5 to 88.2% when the activated samples are heated from 50 to 80 °C. Similarly, the leaching efficiency of the yttrium increased from 4.5 to 46.5% for the inactivated samples. This indicated that a moderately high temperature is required. When the temperature exceeds 80 °C, the leaching efficiency of the yttrium remains nearly unchanged at 12 h. Consequently, 80 °C was chosen as the optimal leaching temperature for the following experiments. With increasing reaction temperatures, the available energy for the activation of atoms and molecules correspondingly increases. Furthermore, as the temperature increased, the mass transfer coefficient and rate of chemical reaction enhanced [23].

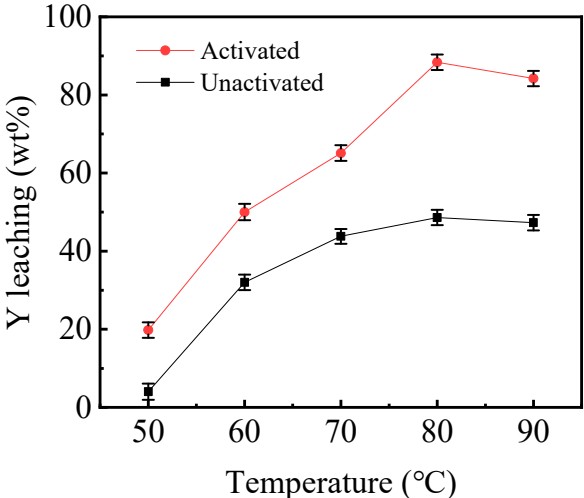

**Figure 6.** The effect of reaction temperature on leaching efficiency before and after mechanical activation ($t_{MA}$ = 60 min t $_L$= 12 h).

### 3.6. Effect of Reaction Time on Leaching Efficiency before and after Mechanical Activation

The effect of the reaction time on the leaching efficiency was explored further, and the results are given in Figure 7. The samples were tested at a variety of reaction temperatures, before and after mechanical activation. When comparing a with b, it was discovered that the metal yttrium in the inactivated sample can hardly be leached out at the reaction temperature of 50 °C, whereas the metal yttrium in the mechanically activated sample was leached in a small amount, and the leaching efficiency tends to be stable as the reaction time increases. When the leaching time was 12 h and the reaction temperature was 80 °C, the leaching efficiency of the mechanically activated samples reached 88.37%, but the leaching efficiency of the inactivated samples was only 48.61%. It was determined that mechanical activation can greatly improve the leaching efficiency of yttrium at higher temperatures, whereas mechanical activation has a minimal influence on the leaching efficiency of metallic yttrium at lower temperatures.

In addition, temperature is known to have a significant effect on the viscosity of DESs. At low temperatures, the viscosity of the $DES_S$ increases, and the diffusion rate of the protons decreases, which makes the reaction between the DESs and the metal yttrium difficult [24]. In contrast, as the temperature of the DESs increased, the viscosity decreased and the mobility of the protons increased. Moreover, at the optimal reaction temperature, mechanical activation increased the energy stored in the sample and the amount of energy that weakens or destroys the crystal structure within the sample. When the kinetic energy was greater than or equal to the activation energy, the number of molecules increased, which led to an increase in the number of effective collisions, thereby accelerating the rate

of chemical reaction [25,26]. Therefore, the yttrium leaching efficiency can be significantly increased at the same time. The result that the mechanical activation increases the reaction rates indicated that the system diffusion was controlled (either externally or through an inert product layer), and the same should be true with the temperature, so actually the region where diffusion is manifested should simultaneously be mechanical activation and high temperature.

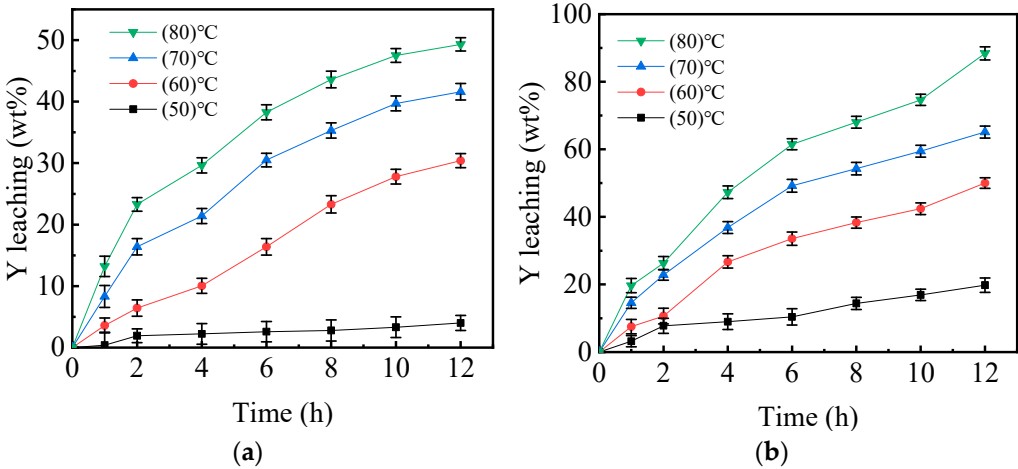

**Figure 7.** Effect of leaching time on the leaching efficiency of metallic yttrium in different activated samples at different reaction temperatures (T = 80 °C, $t_{MA}$ = 60 min). (**a**) Mechanically inactivated samples; (**b**) Mechanically activated samples.

### 3.7. Apparent Reaction Activation Energy

The leaching process of rare-earth metal from $DES_S$ is a homogeneous uncatalyzed reaction in a fluid–solid system; various models have been proposed for exploring the kinetics of this kind of process, including the shrinking-core model, the progressive-conversion model, and the grain model. The progressive-conversion and grain models are focused on describing the leaching kinetics of porous solids [27,28], while both of the inactivated and activated rare-earth carbonates were compact particles with no porous solids, according to the morphology images by the scanning electron microscopy, and the shrinking-core model is adopted for describing the kinetics of rare-earth carbonate decomposition and the leaching process [29].

The leaching behavior of rare-earth carbonates in ChCl–Ur–MA DESs differs from that of conventional alkaline and acid leaching. The yttrium dissolution process can be described in terms of a complex reaction relating to concentrations of surface ligand (Ur and $Cl^-$). During the leaching of DESs, free urea and $Cl^-$ were gradually consumed due to the production of [Y.urea $Cl^-$] complexes. A shrinking core model was developed to describe the dissolving kinetics of Y in the absence of a product layer. The reaction of Y complexation with ligands (Ur and $Cl^-$) and diffusion of ligands or produced complexes are involved in the Y leaching process. Therefore, the integrated rate equation is provided below, considering that the ligand concentrations are nearly constant [30]:

$$1 - (1 - \alpha)^{\frac{1}{3}} = \frac{k'[\text{Ur}]^m [Cl^{-1}]^n}{\rho_Y r_0} t = kt \tag{1}$$

where $\alpha$ is the leaching efficiency of Y at time t; k represents an apparent rate constant; [Ur] and [$Cl^-$] denote the Ur and $Cl^-$ concentrations on the unreacted Y core surface, respectively, which are both unchanged; m and n correspond to the apparent reaction order; $\rho_Y$ represents the Y molar density; $r_0$ represents the particle's initial radius; k denotes the kinetic constant calculated from Equation (1). Figure 8 depicts the plots of $1 - (1 - \alpha)^{1/3}$ versus leaching time (t) at different temperatures. According to the experimental data, it

was evident that the experimental data fit well with the shrinking core model. The higher viscosity of ChCl−Ur−MA DESs results in a slower rate of reactant diffusion (Ur and Cl$^-$).

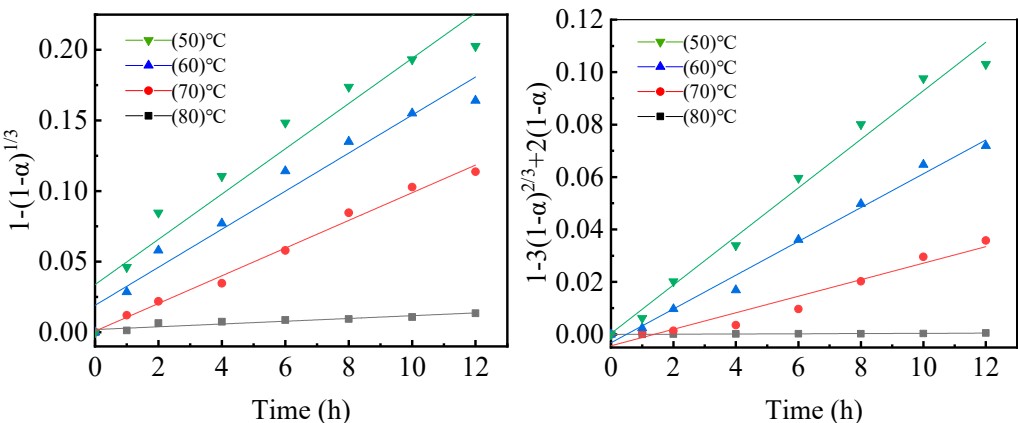

**Figure 8.** Kinetic model fitting curves of inactivated samples at different reaction temperatures.

Furthermore, using the k value obtained by fitting, we can calculate the apparent activation energy of the leaching reaction, based on the Arrhenius equation:

$$lnk = -\frac{Ea}{RT} + lnA \tag{2}$$

where A represents a frequency factor; Ea describes the apparent activation energy of the leaching reaction; $R$ represents a gas constant; and T represents the leaching temperature. The logarithm of the kinetic rate constant (*lnk*) versus 1/T had a good linear relationship (Figure 9a). The apparent activation energy was determined to be 83.88 kJ·mol$^{-1}$ based on the slope. These findings indicated that the chemical reaction controls the leaching process.

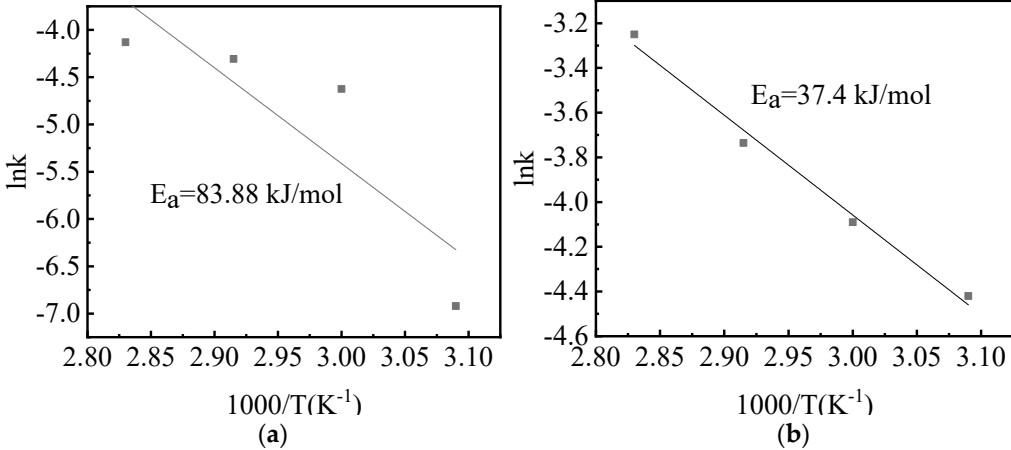

**Figure 9.** Arrhenius diagram of the sample. (**a**) Mechanically inactivated sample; (**b**) Mechanically activated samples.

Using the same analysis method as described above, the plots of $1 - (1 - \alpha)^{1/3}$ and $1 - 3(1 - \alpha)^{2/3} + 2(1 - \alpha)$ versus leaching time, respectively, were tested and are presented in Figure 10. As demonstrated in Figure 10, the linear equation showed good fits with the experimental results. For the plots of $1 - (1 - \alpha)^{1/3}$ versus the leaching time, the minimum value of the adjusted coefficient of determination (adjusted (Adj.) R$^2$) of these plots was 0.9713, while it was 0.9878 for the plots of $1 - 3(1 - \alpha)^{2/3} + 2(1 - \alpha)$ versus the leaching time. These findings suggested that the diffusion step was more likely to control the leaching of Ye from the activated sample.

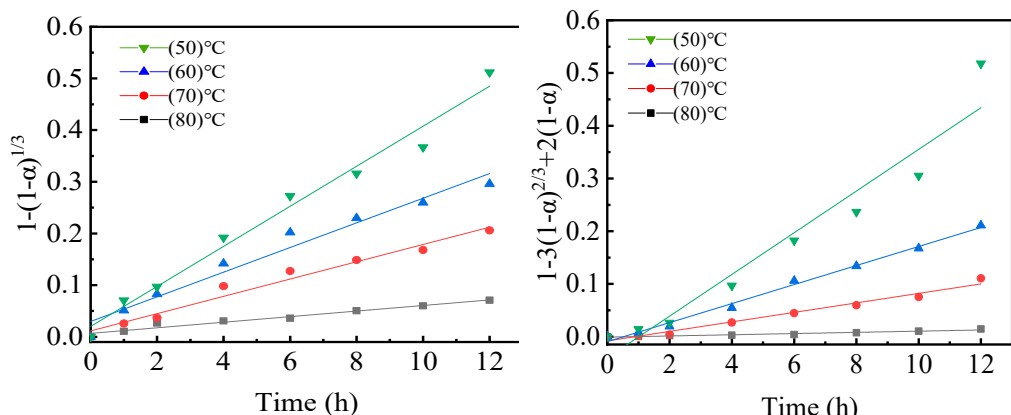

**Figure 10.** Kinetic model fitting curves of mechanically activated samples at different reaction temperatures.

Additionally, the apparent activation energy ($E_{ap}$) of the sample that was mechanically activated for 60 min at 1800 r·min$^{-1}$ ball milling speed was calculated, and the apparent activation energy of the sample was reduced to 37.4 kJ·mol$^{-1}$ after the mechanical activation pretreatment. This result further indicates that the leaching process is controlled by the diffusion through a liquid phase boundary layer.

The above study suggested that the apparent activation energy of the sample can be significantly reduced by the method of the mechanical activation pretreatment, the dependence of the sample on the temperature during the reaction process can be reduced, and the leaching process was changed from controlled by chemical reaction to controlled by diffusion in the solid product layer.

## 4. Conclusions

In this work, the leaching behavior of rare-earth carbonate before and after mechanically activated samples in ChCl–urea–MA DESs was investigated. The optimal conditions for the leaching of rare-earth carbonate were a leaching temperature of 80 °C and a leaching time of 12 h. Under the optimal conditions, the inactivated sample had a leaching efficiency of 48.61%, while the mechanically activated sample had a leaching efficiency of 88.37%, indicating that the mechanical activation pretreatment is an effective method for improving the leaching efficiency of yttrium in rare-earth carbonate. The chemical reaction controlled the leaching process of the inactivated samples at temperatures ranging from 50 to 80 °C, and the apparent activation energy was 83.88 kJ·mol$^{-1}$. However, the apparent activation energy value decreased to 37.4 kJ·mol$^{-1}$ after mechanical activation for 60 min at 1800 r·min$^{-1}$, which was controlled by diffusion. It can be seen that the mechanical activation pretreatment can significantly reduce the dependence of the rare-earth carbonate-leaching reaction on the reaction temperature, and the leaching process of the sample was changed from a chemical reaction to diffusion in the solid product layer.

**Author Contributions:** Writing—original draft, X.L., W.L., Y.G. and G.T. All authors have read and agreed to the published version of the manuscript.

**Funding:** This work was supported by the National Natural Science Foundation of China (No. 21665027), and program for the Applied Basic Research Project of Yunnan Province Youth Program (No. 2017FD118), and Scientific research fund project of Yunnan Education Department (No. 2022J0443).

**Institutional Review Board Statement:** Not applicable.

**Informed Consent Statement:** Not applicable.

**Data Availability Statement:** Not applicable.

**Conflicts of Interest:** The authors declare no conflict of interest.

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
