# Peer review of "Effect of Mechanical Activation on the Leaching Process of Rare Earth Metal Yttrium in Deep Eutectic Solvents"

_applsci, doi:10.3390/app12168253_

Round 1
Reviewer 1 Report
The topic is novel and of great interest. However, I have some suggestions oriented to improve the quality of the presentation:
1.- I recommend to review the methodology section, specifically section 2.4.
2.- It would be interesting to add a UV-Vis pattern where the characteristic peaks of the formed complex were observed. This, because the UV-Vis is the technique to determine the leaching %.
3.- It would be interesting to present a micrography where the morphology changes can be observed. This can support the proposed kinetic mechanism.
Reviewer 2 Report
Generally I do not recommend publication in the present form Two optios:
1)shorten it to a note or
2) After major revisions including XRD and/or possible FT-IR otherwise conclusions on mechanisms are ungrounded
3) Model needs further explanation regarding transference of shrinking-core reaction models to the present situation
4) Error bars in data reporting are needed given the easiness of the type of chemical analysis used (i.e UV measurements)
Please see attached file

Round 2
Reviewer 2 Report
It has been improved but still needs major revisions of paragraphs 3.6 and 3.7 please see detailed comments in the attached file

Author Response
Dear editor:
Thank you for your letter. The paper manuscript entitled "Effect of mechanical activation on the leaching process of rare earth metal yttrium in deep eutectic solvents”has been revised according to the reviewers' opinion. The response to the reviewers are as follows:
1) In introduction: “In general, it is not difficult to dissolve metal oxides in molecular solvents…” I think it’s the opposite please check “not” should be deleted. I have pointed this out in my previous review.
Answer:“not” has been deleted in introduction.
2) In Figures 3,4,5 and 6 the values of the parameters kept constant should be indicated on the figures not just in the text.
Answer:To keep the diagrams simple and clear, the parameter values that remain the same have been added and highlighted on the figure caption.
3) In fig 6 the leaching time is required (is it 12h? and constant? As discussed in the text?) it should be placed both on the figure caption and in the text.
Answer:the leaching time has been added ( tL= 12 h) both on the figure caption
and in the text.
4) Figure 5 title it reads: “The effect of mechanical activation on leaching time and leaching efficiency” should be “The effect of mechanical activation and leaching time on leaching efficiency”
Answer:The sentence has been revised and highlighted in red.
5) Figure 4 caption reads: “ The effects of different mechanical activating times on Y leaching efficiency” it should be “The effect of mechanical activation time on leaching efficiency”
Answer:The sentence has been revised and highlighted in red.
6) Figure 7 caption: Is reaction time the same as leaching time? Ι think leaching time should be used to be consistent with previous terminology. Following in page 6: “The influence of leaching temperature on the yttrium… ……………….. of yttrium is positively correlated with temperature.” I really think the whole paragraph should be omitted. It goes back to figure 6 and does not add anything new.
Answer:Reaction time is leaching time, and it has been uniformly revised to
leaching time. And the whole paragraph has been omitted.
7) Following: “The main reason is that the temperature has………..” I think it should start as “In addition temperature is known to have a significant effect on viscosity……….” But also the whole paragraph should be rewritten… the English is a bit cumbersome which is understandable, nevertheless it should be written more clearly. It is very difficult from this paragraph to understand if the system is reaction or diffusion controlled. I thing it’s diffusion controlled but it is not articulated clearly by the authors. Certainly viscosity has an influence particularly on diffusion rate of protons. Below that however, discussion starts on factors affecting reaction rates. From what I understand two parameters i.e. mechanical activation time and temperature are examined by the authors. Now mechanical activation certainly as it is argued increases reaction rates which should then make the system diffusion controlled (either external or through an inert product layer) the same should be true with temperature so actually the region where diffusion is manifested should be mechanical activation and high temperature simultaneously. Now if temperature is raised however, the viscosity decreases and diffusion rates are increased as explained by the authors which should make the system reaction controlled so the region where reaction dominance is manifested should be high temperatures and no mechanical activation. So the authors should separate their data between these two extremes although temperature enhances both reaction and diffusion rates. So the system is quite complicated.
Overall particularly this paragraph is very important but it should a) moved after the presentation of the kinetic model and b) be rephrased so that the meaning is articulated more clearly.
Answer:This paragraph has been moved after the presentation of the kinetic model, and it has been rephrased and the meaning is articulated more clearly.
8) The same is true for section 3.7 i.e. it should be presented more clearly according to the above. Maybe the authors should examine more carefully their data on this aspect i.e. diffusion or reaction controlled regions. The apparent activation energies should be given 2 with a standard error. In the case of 9a there is great inaccuracy in the estimation of Ea it seems to me that one cannot estimate Ea since the mechanism is mixed (i.e. both reaction and diffusion rates are significant and this is manifested in the scatter of fig.9a). Also Figure 9 of the activation energies should be placed after fig.10 (for mechanically activated samples). Looking again in Figs 10 and 8 it seems to me that diffusion dominates at high temperatures irrespective of mechanical activation. Diffusion becomes more evident in mechanically activated samples. The authors should concentrate on these higher temperatures and calculate the respective apparent activation energies for activated and inactivated samples in this higher temperature regime where scatter is less and accuracy for Ea will be greater.
Answer:
9) In paragraph 3.7 it reads: “homogeneous uncatalyzed reaction” should be “heterogeneous non-catalytic reaction system” ( we still have two phases present)
Overall I don’t think that in inactivated samples reaction dominates. We can not say that. It seems to me that even in samples not mechanically activated, diffusion (through inert product) dominates at high temperatures. The least that can be said is that in inactivated samples a mixed mechanism is present. So conclusions should change accordingly. Very important: the paragraphs 3.6 and 3.7 should be rewritten and revised substantially according to above considerations and the concepts be conveyed more clearly because they constitute the critical part of the paper.
Answer:
